P3H-25-079, TTK-25-24, IPPP/25/69

# Threshold improved $ZH$ production at the LHC

**Goutam Das**[1⋆], **Chinmoy Dey**[2,3†], **M. C. Kumar**[2‡], **and Kajal Samanta**[4∘]

**1** Institut für Theoretische Teilchenphysik und Kosmologie, RWTH Aachen University,
D-52056 Aachen, Germany
**2** Department of Physics, Indian Institute of Technology Guwahati,
Guwahati-781039, Assam, India
**3** Theoretical Physics Division, Physical Research Laboratory,
Navrangpura, Ahmedabad 380009, India
**4** Institute for Particle Physics Phenomenology, Durham University,
Durham DH1 3LE, United Kingdom

⋆ goutam@physik.rwth-aachen.de , † d.chinmoy@iitg.ac.in , ‡ mckumar@iitg.ac.in ,
∘ kajal.samanta@durham.ac.uk

## Abstract

**We present precise theoretical results for the $ZH$ production cross section and invariant mass distribution at the Large Hadron Collider (LHC) taking into account the effects of soft gluons. We improve both quark-initiated and gluon-initiated subprocesses through threshold resummation within the QCD framework and present combined results relevant for 13.6 TeV LHC.**

# 1 Introduction

The associated production of the Higgs boson with a $Z$ boson at the Large Hadron Collider (LHC) plays a crucial role in probing the Higgs couplings to the electroweak gauge bosons. Both the ATLAS and CMS collaborations are actively pursuing precision measurements of this process [1–4]. To fully exploit the experimental precision, equally precise theoretical predictions are essential. The dominant production mode for $ZH$ in LHC is through the Drell-Yan (DY) type quark-antiquark annihilation ($q\bar{q} \rightarrow Z^* \rightarrow ZH$) for which the higher order Quantum Chromodynamics (QCD) correction has been known to next-to-next-to-next-to-leading order (N3LO) [5] accuracy. In addition to the DY-type contribution, other subprocesses also contribute to $ZH$ production, such as bottom quark annihilation [6] and top quark loop–induced contribution [7]. However, these contributions are subleading, typically contributing at the sub-percent level relative to the NLO QCD DY-type prediction. Electroweak (EW) corrections to the DY-type channel have also been computed at NLO [8], yielding a negative correction of approximately 5% relative to the NLO QCD result. Starting from N2LO, the $ZH$ production receives contributions from the gluon fusion channel. Although this subprocess is suppressed by two powers of the strong coupling constant ($\alpha_s$) relative to the DY-type channel, the suppression is largely offset by the substantial gluon luminosity at the LHC, leading to a significant overall contribution.

Given its growing importance, the gluon fusion subprocess ($ggZH$) has been extensively studied in the literature at both leading order (LO) [9–12] and next-to-leading order (NLO) [13–21]. At NLO, the total cross section of this channel roughly doubles compared to the prediction of LO, while the theoretical uncertainty due to renormalization and factorization scale variations is reduced to about 15%. The fixed order results thus suffer from the large threshold logarithms arising from soft gluons emission. Indeed, the threshold soft-virtual (SV) logarithms at NLO can contribute to $90 - 99\%$ of the complete NLO results in the range $Q = 350 - 2000$ GeV [1]. By resumming these SV logarithms to all orders, one obtains predictions that are stable and well-behaved across the relevant kinematic regions. The formalism for threshold resummation is well established in the literature [24–35], and has been successfully applied to improve theoretical predictions for both inclusive cross-sections and differential observables such as invariant mass distributions.

In this report, we present an improved theoretical description of the $ZH$ process at LHC by incorporating threshold resummation effects at the SV level for gluon fusion as well as for the DY-type channels to $ZH$ production. Our analysis [36] covers both the total cross-section and the invariant mass distribution of the $ZH$ pair. We employ the Born-improved gluon fusion framework, which has proven effective in the case of inclusive Higgs production, and we expect it to yield similarly reliable results for the $ZH$ channel. The article is organized as follows: In Section 2, we introduce the key theoretical formulas. In Section 3, we provide a phenomenological study for the gluon fusion subprocess, combining it with DY-type contributions to present complete results for $pp$ collisions with $\mathcal{O}(\alpha_s^3)$ accuracy. Finally, we conclude in Section 4.

# 2 $ZH$ production at the LHC

The hadronic cross-section for $ZH$ production at LHC can be written as,

---

[1]Similar observation has also been made for the DY [22] and Higgs [23] processes.

$$Q^2 \frac{\mathrm{d}\sigma}{\mathrm{d}Q^2} = \sum_{a,b} \int_0^1 \mathrm{d}x_1 \int_0^1 \mathrm{d}x_2 \, f_a(x_1, \mu_F^2) f_b(x_2, \mu_F^2) \int_0^1 \mathrm{d}z \, \delta(\tau - zx_1x_2) Q^2 \frac{\mathrm{d}\widehat{\sigma}_{ab}(z, \mu_F^2)}{\mathrm{d}Q^2}, \quad (1)$$

where $f_{a,b}$ are the parton distribution functions (PDFs) for parton $a$, $b$ in the incoming protons and $\widehat{\sigma}_{ab}$ is the partonic coefficient function. The hadronic and partonic threshold variables $\tau = Q^2/S$ and $z = Q^2/\widehat{s}$ are defined in terms of respective center-of-mass energies $S$ and $\widehat{s}$. Here $Q$ is the invariant mass of the $ZH$ system and $\mu_F$ is the factorization scale. The partonic coefficient function can be decomposed in soft-virtual ($\Delta_{ab}^{\mathrm{SV}}$) and regular ($\Delta_{ab}^{\mathrm{REG}}$) parts at each order in $\alpha_S$. The SV part contains all dominant singular contribution in the limit $z \to 1$ whereas the regular part contains sub-dominant contributions. The SV logarithms can be resummed in Mellin-$N$ space and corresponding resummed partonic coefficient takes the form,

$$\frac{1}{\widehat{\sigma}_{ab}^{(0)}(Q^2)} Q^2 \frac{\mathrm{d}\widehat{\sigma}_{N,ab}^{\mathrm{N}n\mathrm{LL}}}{\mathrm{d}Q^2} = \int_0^1 \mathrm{d}z \, z^{N-1} \Delta_{ab}^{\mathrm{SV}}(z) \equiv g_0(Q^2) \exp\left(G_N^{\mathrm{SV}}\right). \quad (2)$$

The function $G_N^{\mathrm{SV}}$ contains the universal threshold exponent and determines the resummed accuracy through its expansion (see for example in [37–39]). The constant $g_0(Q^2)$ contains the process-dependent information (see [36, 39]). Using *minimal prescription* [40] for Mellin inversion, one can finally find resummed results in the physical $z$-space which can be also matched to the corresponding fixed order results to incorporate missing regular corrections,

$$Q^2 \frac{\mathrm{d}\sigma_{ab}^{\mathrm{N}n\mathrm{LO}+\mathrm{N}n\mathrm{LL}}}{\mathrm{d}Q^2} = Q^2 \frac{\mathrm{d}\sigma_{ab}^{\mathrm{N}n\mathrm{LO}}}{\mathrm{d}Q^2} + \sum_{ab \in \{gg, q\bar{q}\}} \widehat{\sigma}_{ab}^{(0)}(Q^2) \int_{c-i\infty}^{c+i\infty} \frac{\mathrm{d}N}{2\pi i} \tau^{-N} f_{a,N}(\mu_F) f_{b,N}(\mu_F)$$

$$\times \left(Q^2 \frac{\mathrm{d}\widehat{\sigma}_{N,ab}^{\mathrm{N}n\mathrm{LL}}}{\mathrm{d}Q^2} - Q^2 \frac{\mathrm{d}\widehat{\sigma}_{N,ab}^{\mathrm{N}n\mathrm{LL}}}{\mathrm{d}Q^2}\bigg|_{\mathrm{tr}}\right). \quad (3)$$

Other prescriptions, for instance the Borel prescription [41–43], may lead to differences with respect to the minimal prescription that are confined to subleading terms.

## 3 Results

All the ingredients necessary for performing soft-gluon resummation in both the gluon fusion and Drell–Yan (DY)–type subprocesses for $ZH$ production are available in [36, 39], and can be used to quantitatively assess their impact at 13.6 TeV LHC. Our choice of parameters are given below:

$$\sqrt{S} = 13.6 \text{ TeV}, \quad \mathrm{PDF} = \mathrm{PDF4LHC21\_40} \, [44], \quad \alpha_S(m_Z) = 0.1180,$$
$$\alpha \simeq 1/127.93, \quad m_Z = 91.1880 \text{ GeV}, \quad \Gamma_Z = 2.4955 \text{ GeV},$$
$$m_W = 80.3692 \text{ GeV}, \quad m_t = 172.57 \text{ GeV}, \quad m_H = 125.2 \text{ GeV}. \quad (4)$$

The weak mixing angle is then determined by $\sin^2\theta_{\mathrm{w}} = (1 - m_W^2/m_Z^2)$ corresponding to the Fermi constant $G_F \simeq 1.2043993808 \times 10^{-5} \text{ GeV}^{-2}$. To account for different types of uncertainties, we have used the following formulas;

$$\delta(\mathrm{PDF}) = \left(\sum_{i=1}^{40} (\sigma(i) - \sigma(0))^2\right)^{1/2}, \quad (5a)$$

$$\delta(\alpha_s) = \left|\frac{3}{4} \frac{\sigma(\alpha_s^c(m_Z) = 0.119) - \sigma(\alpha_s^c(m_Z) = 0.117)}{\sigma(\alpha_s^c(m_Z) = 0.118)}\right|, \quad (5b)$$

$$\delta(\mathrm{PDF} + \alpha_s) = \sqrt{\delta(\alpha_s)^2 + \delta(\mathrm{PDF})^2}. \quad (5c)$$

For the scale uncertainty, we use seven-point scale variation where we vary both $\mu_R$ and $\mu_F$ by factor 2 or 1/2 around the central value with constraints $|\ln(\mu_R/\mu_F)| \leq \ln(2)$.

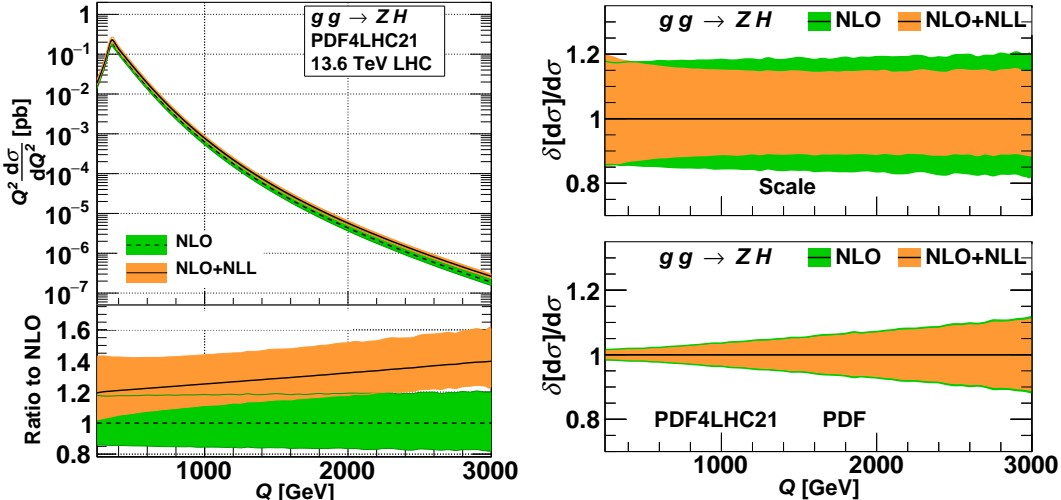

Figure 1: The contribution from $gg$ subprocess is shown for fixed order and resummed cases with the corresponding uncertainties.

The fixed-order results for DY and gluon fusion channel have been computed using the publicly available code n3loxs [5] and vh@nnlo [13,45,46] respectively. In the left-top panel of Fig. 1, we present contributions for gluon fusion channels [36] along with the seven-point scale uncertainties around central scales $(\mu_R^c, \mu_F^c) = (Q, Q)$. To quantify the enhancement due to resummation, we take the ratio with respect to NLO which are shown in the left-bottom panel of Fig. 1. The enhancement due to resummation in the higher-$Q$ region is about 40% for $gg$ subprocess compared to respective NLO $ggZH$ result. There is also a significant reduction in the scale uncertainty after inclusion of the threshold effects, by 5.0%. In the right-top panel of Fig. 1, we show such a comparison for the scale uncertainty of $gg$ subprocess at NLO and NLO+NLL. In the right bottom panel, we present the intrinsic PDF uncertainty for both NLO and NLO+NLL which stays similar for both fixed order and resummed case and amounts to below 2% in the low-invariant mass region.

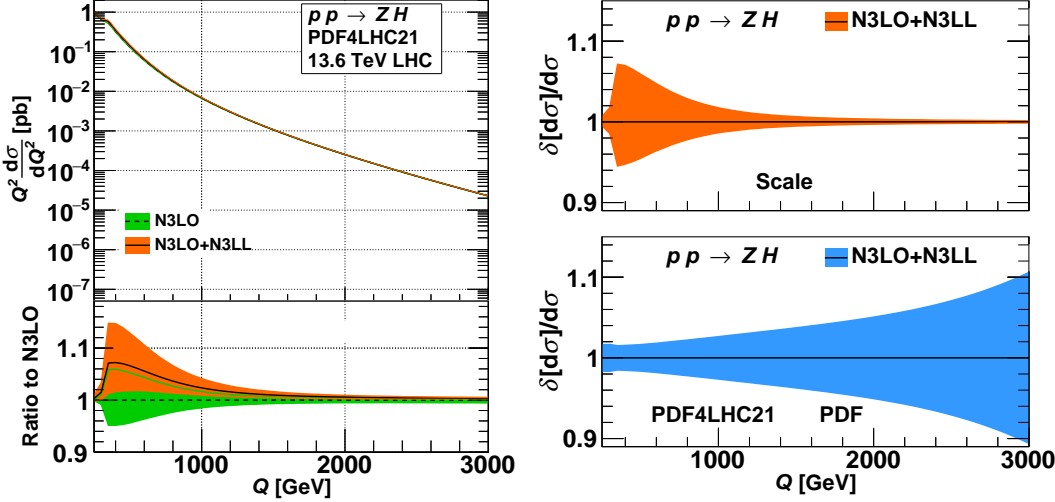

Figure 2: The total contributions for the $pp \to ZH$ process (combination of DY-type and $gg$ subprocess) and the corresponding uncertainties.

For completeness, we combine the gluon fusion and DY-type contributions to obtain the full $pp \to ZH$ results at $\mathcal{O}(\alpha_s^3)$ level. Particularly, for the fixed order case, we combine the N3LO DY contribution to NLO $gg$ contribution to have total N3LO results and in the resummation case, we combine the N3LL DY [39] contribution to NLL $gg$ contribution to have total N3LO+N3LL results. These combined results are shown in Fig. 2. The right panel of Fig. 2 reflects the corresponding seven-point scale uncertainty and intrinsic PDF uncertainty. We observe that the scale uncertainty remains below 10% around $ZH$ threshold, while the PDF uncertainty stays below 2% in the same region.

| Order | Central (fb) | $\Delta$(RESUM) | $\delta$(Scale) | $\delta$(PDF) | $\delta(\alpha_s)$ | $\delta$(PDF + $\alpha_s$) |
|---|---|---|---|---|---|---|
| $\sigma_{gg}^{\text{NLO+NLL}}$ | 151.3 | 21.29% | ±19.4% | ±0.7% | ±2.3% | ±2.4% |
| $\sigma_{\text{DY}}^{\text{N3LO+N3LL}}$ | 841.6 | 0.01% | ±0.6% | ±0.8% | ±0.8% | ±1.1% |
| $\sigma_{tot}^{\text{N3LO+N3LL}}$ | 1004.2 | 2.72% | ±3.0% | ±0.6% | ±1.0% | ±1.2% |

Table 1: Inclusive resummed cross-sections for $ZH$ are presented (for $gg$, DY type and, total) for 13.6 TeV with scale, PDF and $\alpha_s$ uncertainties. $\sigma_{tot}^{\text{N3LO+N3LL}}$ contains both fixed order and resummed results from $gg$ and DY-type as well as contributions from top-loop and bottom annihilation channels at fixed order.

Finally, in Table 1, we present the total cross-sections including resummation effects from $gg$, DY-type and their combined contribution ($tot$). The later contains, not only the $gg$ and DY-type resummation effects, but also all fixed order results including $gg$ at NLO, DY-type at N3LO, contributions from bottom quark annihilation [6], top-loop–induced processes [7], where the Higgs boson is radiated from a closed top-quark loop. We present the enhancement due to resummation over fixed order through $\Delta(\text{RESUM}) = (\text{RESUM} - \text{FO})/\text{FO} \times 100\%$. Additionally, we report the associated theoretical uncertainties: the seven-point scale variation, intrinsic PDF uncertainties, and those arising from the strong coupling constant. For the later, we use a $1\sigma$ variation $\left(\alpha_S^{\pm}(m_Z) = \alpha_S^c(m_Z) \pm 0.0015\right)$ of strong coupling around its central value $\alpha_S^c(m_Z) = 0.118$. The cross-section for $\alpha_S^{\pm}(m_Z)$ has been computed from the subsets $41, 42$ according to PDF4LHC recommendation [44]. The combined PDF+$\alpha_S$ uncertainty is obtained by adding the individual contributions in quadrature. We observe that the largest source of uncertainty arises from the scale variation in gluon fusion channel, which remains sizable at about 19% at NLO+NLL level, indicating the need for further improvements through higher-order computation.

# 4   Conclusion

To summarize, we have studied the impact of soft gluon resummation on $ZH$ production at the LHC, focusing in particular on the gluon fusion subprocess. For this channel, we employed the Born-improved NLO framework and matched it with next-to-leading logarithmic (NLL) resummed results. Our analysis shows that soft-virtual (SV) resummation yields an additional enhancement of approximately $20 - 40\%$ over the fixed-order NLO prediction in the kinematic region considered. Furthermore, the seven-point scale uncertainty is reduced by about 5% compared to the NLO result, with the most notable improvement occurring in the high-$Q$ regime. We have also assessed the uncertainty associated with PDFs in the resummed predictions, finding it to be below 2% in the low invariant mass region.

Finally, to provide experimentally relevant predictions, we combine contributions from all

relevant subprocesses—including soft gluon resummation effects in both the $ggZH$ and DY-type channels, and present results for the invariant mass distribution and total production cross section at the 13.6 TeV LHC.

**Funding information** This research has been supported by the Deutsche Forschungsgemeinschaft (DFG, German Research Foundation) under grant 396021762 - TRR 257 (*Particle Physics Phenomenology after Higgs discovery.*), the SERB Core Research Grant (CRG) under the project CRG/2021/005270, the Royal Society (URF/R/231031) and the STFC (ST/X003167/1 and ST/X000745/1).

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
