# Peer review of "Threshold improved $Z H$ production at the LHC"

_SciPost Physics Community Reports, doi:SciPost Phys. Comm. Rep. 14 (2025)_

## Round 1 · Referee Report · Anonymous (Referee 2) · 2025-10-17

Strengths

  1. Calculations for an important Standard Model process, ZH production.
  2. Results are given at high perturbative orders.

Weaknesses

  1. The resummed results are prescription dependent.
  2. No discussion is given on the relative contributions of soft-gluon corrections at orders were the complete results are known.

Report

The submitted manuscript discusses soft-gluon corrections for ZH production at the LHC. The authors present cross sections at 13.6 TeV LHC energy at NLO and NLO+NLL for the gg subprocess, and at N3LO and N3LO+N3LL for the DY subprocess. ZH production is an important process, and it is essential to have accurate predictions for it.

The authors write that "The fixed order results thus suffer from the large threshold logarithms arising from soft gluons emission." Yet, they do not explicitly provide any numbers to show the percentage contribution of these logarithms to the known complete results through N3LO. It would be nice to have a short discussion on that.

The authors also state that "To obtain stable and accurate predictions, it is essential to resum these large soft-virtual (SV) logarithms to all orders." This is an oft-repeated but largely incorrect and meaningless statement. Numerous studies have shown that expansions of the resummed cross section provide quickly diminishing numbers at higher orders and, in fact, the differences in numerical results among various resummation prescriptions are often larger than contributions beyond N3LO (or even beyond NNLO). Hence, fixed-order expansions to high orders (at least one order higher than the exact results) is perfectly fine and is often more accurate.

In particular, the authors use the minimal prescription of their Ref. [38]. This prescription, while previously used by the authors and some other groups, has made seriously wrong predictions for the size of higher-order corrections (beyond NLO) for several processes. The NNLO soft-gluon corrections in [38] were more than an order of magnitude smaller than the exact NNLO corrections that were calculated much later. These matters, i.e. differences among prescriptions and, in particular, critiques of the minimal prescription, have been studied in numerous papers since then, and the authors should acknowledge the prescription dependence of their results.

Requested changes

  1. Provide some discussion of the contribution of the soft-gluon logarithms at fixed order (either percentage or actual numbers).
  2. Delete or modify the sentence about "essential to resum ... to all orders".
  3. State that the results are prescription dependent.

Recommendation

Ask for minor revision

  • validity: ok
  • significance: good
  • originality: good
  • clarity: high
  • formatting: excellent
  • grammar: excellent

Author:  Kajal Samanta  on 2025-10-31  [id 5978]

(in reply to Report 2 on 2025-10-17)

Dear Editor,
We thank the referee for providing feedback. Please find below our response.

"The authors write that ‘The fixed order results thus suffer from the large threshold logarithms arising from soft gluons emission.’ Yet, they do not explicitly provide any numbers to show the percentage contribution of these logarithms to the known complete results through N3LO. It would be nice to have a short discussion on that. The authors also state that ‘To obtain stable and accurate predictions, it is essential to resum these large soft-virtual (SV) logarithms to all orders.’ This is an oft-repeated but largely incorrect and meaningless statement. Numerous studies have shown that expansions of the resummed cross section provide quickly diminishing numbers at higher orders and, in fact, the differences in numerical results among various resummation prescriptions are often larger than contributions beyond N3LO (or even beyond NNLO). Hence, fixed-order expansions to high orders (at least one order higher than the exact results) is perfectly fine and is often more accurate."

Resummation plays a crucial role in collider phenomenology, providing reliable theoretical predictions where fixed-order calculations alone are insufficient. While fixed-order results offer valuable theoretical insights, they often fail to accurately describe experimental data unless computed to very high orders, which is typically impractical. Moreover, for many observables (such as transverse momentum distributions), even higher-order predictions (e.g. up to N3LO) may not adequately capture the correct behavior in certain kinematic regions, such as the low-pT regime. This limitation has been well established in numerous studies over the past several decades. In the context of threshold resummation for colorless final states, e.g. the Higgs boson production at the LHC, as given in Fig.2 and Table-3 of [1], the fixed order results change from 36.9 pb at NLO to 46.5 pb at NNLO for the scale choice of $\mu_0=m_H/2$, while the resummation prediction at NLO+NLL is 46.2 pb, making the resummation contribution quite sizeable. Similarly, the NNLL contributions are non-negligible compared to the $\mathcal{O}(\alpha_s^3) $ corrections. Thus, in the low Q region, due to the large parton fluxes, the resummation effects are non-negligible. For the case of $q\bar{q}$ initiated DY type ZH production process, in the high invariant mass region, see. Fig.8 of [2], it is shown that the N3LO results are about 30% of LO, while the resummation at NNLO+NNLL contributes about 31%. It is true that the difference between NLO+NLL and NNLO+NNLL is smaller compared to the one between NNLO and N3LO in the high invariant mass region, say Q = 3000 GeV. However, there is a reduction in the scale uncertainties as a result of resummation in the same Q-region, see the left panel of Fig.16 of [2]. For threshold resummation, the impact is generally less dramatic but still significant, particularly in improving fixed-order predictions when higher-order results are challenging to obtain, as is the case for gluon-fusion-induced ZH production discussed in the current work.

"In particular, the authors use the minimal prescription of their Ref. [38]. This prescription, while previously used by the authors and some other groups, has made seriously wrong predictions for the size of higher-order corrections (beyond NLO) for several processes. The NNLO soft-gluon corrections in [38] were more than an order of magnitude smaller than the exact NNLO corrections that were calculated much later. These matters, i.e. differences among prescriptions and, in particular, critiques of the minimal prescription, have been studied in numerous papers since then, and the authors should acknowledge the prescription dependence of their results. "

As with any theoretical framework (including fixed-order perturbation theory), resummation procedures may involve additional prescriptions. These can be viewed as tools to explore theoretical uncertainties rather than as drawbacks. The minimal prescription is one such widely-used standard approach, and alternative schemes (such as the Borel prescription) are also available, and their predictions differ only by subleading terms, which may nonetheless be relevant in specific regions of the distributions. Having said that, the requested minor changes are reasonable for us.

"1. Provide some discussion of the contribution of the soft-gluon logarithms at fixed order (either percentage or actual numbers)."

We have added the following sentence highlighting the contributions from SV logarithms on page-2, in the introduction:
Indeed, the threshold soft-virtual (SV) logarithms at NLO can contribute to 90 − 99% of the complete NLO results in the range Q = 350 − 2000 GeV.

"2. Delete or modify the sentence about ‘essential to resum ... to all orders’. "

We have rephrased the sentence accordingly on page-2, second paragraph: By resumming these soft-virtual (SV) logarithms to all orders, one obtains predictions that are stable and well-behaved across the relevant kinematic regions.

"3. State that the results are prescription dependent. "

We have added a sentence on this on page-3, after eq.(3): Other prescriptions, for instance the Borel prescription, may lead to differences with respect to the minimal prescription that are confined to subleading terms.

Sincerely,
Authors.

References
[1] Marco Bonvini, Simone Marzani, Claudio Muselli, and Luca Rottoli. On the Higgs cross
section at N3LO+N3LL and its uncertainty. JHEP, 08:105, 2016.

[2] Goutam Das, Chinmoy Dey, M. C. Kumar, and Kajal Samanta. Threshold enhanced cross
sections for colorless productions. Phys. Rev. D, 107(3):034038, 2023.

---

## Round 1 · Referee Report · Anonymous (Referee 1) · 2025-10-17

Strengths

Good summary of the status of the effects of soft-gluon resummations applied to the gluon-fusion channel of ZH production.

Weaknesses

It would be interesting to investigate the uncertainties due to the scheme and scale of top quark renormalisation, which are the dominant source of uncertainty in other top-quark-mediated gluon fusion di-boson amplitudes.

Report

The manuscript presents a brief overview of the effect of soft-gluon resummation on the gluon-fusion channel which contributes to ZH production. Due to the gluon luminosity at hadron colliders this channel has a significant contribution, despite being formally NNLO compared to the quark induced Drell-Yann type contributions.

The resummation introduces a substantial shift in the NLO result, particularly in the higher-energy tail, and to some extent reduces the rather large scale uncertainty which is present at NLO.

The combined effect of the resummation of the NNLO ggZH amplitude and N3LO DY amplitude is shown. In the table one can see that the resummation effect of the gluon-fusion channel is significant even at the level of the total N3LO cross section.

Requested changes

None

Recommendation

Publish (easily meets expectations and criteria for this Journal; among top 50%)

---

## Round 2 · List of Changes

"1. Provide some discussion of the contribution of the soft-gluon logarithms at fixed order (either percentage or actual numbers)."
We have added the following sentence highlighting the contributions from SV logarithms on page-2, in the introduction:
Indeed, the threshold soft-virtual (SV) logarithms at NLO can contribute to 90 − 99% of the complete NLO results in the range Q = 350 − 2000 GeV.
"2. Delete or modify the sentence about ‘essential to resum ... to all orders’. "
We have rephrased the sentence accordingly on page-2, second paragraph: By resumming these soft-virtual (SV) logarithms to all orders, one obtains predictions that are stable and well-behaved across the relevant kinematic regions.
"3. State that the results are prescription dependent. "
We have added a sentence on this on page-3, after eq.(3): Other prescriptions, for instance the Borel prescription, may lead to differences with respect to the minimal prescription that are confined to subleading terms.
We have added the following sentence highlighting the contributions from SV logarithms on page-2, in the introduction:
Indeed, the threshold soft-virtual (SV) logarithms at NLO can contribute to 90 − 99% of the complete NLO results in the range Q = 350 − 2000 GeV.
"2. Delete or modify the sentence about ‘essential to resum ... to all orders’. "
We have rephrased the sentence accordingly on page-2, second paragraph: By resumming these soft-virtual (SV) logarithms to all orders, one obtains predictions that are stable and well-behaved across the relevant kinematic regions.
"3. State that the results are prescription dependent. "
We have added a sentence on this on page-3, after eq.(3): Other prescriptions, for instance the Borel prescription, may lead to differences with respect to the minimal prescription that are confined to subleading terms.

---

## Round 3 · Author Response

We have added the references as recommended by the Editor-in-Charge.

---

## Round 3 · List of Changes

Added a footnote on page 2.
Added references [41-43] for the Borel prescriptions.
Added references [41-43] for the Borel prescriptions.

---

## Editorial Decision

published